# The Spatial and Temporal Evolution of Ecological Environment Quality in Karst Ecologically Fragile Areas Driven by Poverty Alleviation Resettlement

**Zhongfa Zhou** [1,2,*], **Qing Feng** [1,2,3], **Changli Zhu** [1,2], **Wanlin Luo** [4], **Lingyu Wang** [1,2], **Xin Zhao** [1,2] **and Lu Zhang** [1,2]

1   School of Karst Science, Guizhou Normal University, Guiyang 550001, China; 20030170040@gznu.edu.cn (Q.F.); 18030090025@gznu.edu.cn (C.Z.); 18030170034@gznu.edu.cn (L.W.); 21030090039@gznu.edu.cn (X.Z.); sophiazl@gznu.edu.cn (L.Z.)
2   National Engineering Research Center for Karst Rocky Desertification Control, Guiyang 550001, China
3   College of Tourism and Aviation Culture, Guizhou City Vocational College, Guiyang 550046, China
4   The Engineering Branch of the Third Institute of Surveying and Mapping of Guizhou Province, Guiyang 550001, China; 19010170430@gznu.edu.cn
*   Correspondence: fa6897@gznu.edu.cn

**Abstract:** Many scholars are skeptical about the poverty reduction effect and the ecological effect of poverty alleviation resettlement (PAR). This study evaluates the spatial and temporal evolution of the ecological environment quality (EEQ) to analyze the effectiveness of ecological restoration from PAR. Based on cloud computing using the Google Earth Engine platform, remote-sensing data were obtained and reconstructed from 2000 to 2020. The remote-sensing ecological index (RSEI) was used to analyze the spatial and temporal evolution of EEQ. The results show that the RSEI of the study area increased by 13.07% after the implementation of PAR, and the rate of increase was higher than that in the period before PAR; the Pu'an and Qinglong areas improved most obviously, in terms of the fragile ecological environment and the prominent contradiction between peasants and land. The residual trends method indicated that the contribution rate of improvement in RSEI due to PAR was 70.56%, 88.38%, and 82.96% in 2017, 2018, and 2020, respectively. An increase in RSEI was more obvious in the area with a greater relocated population and a higher corresponding coupling coordination level. PAR has a promoting effect on EEQ improvement but does not have ecological restoration benefits in every region. It is not satisfactory in terms of the degeneration of the LST indicator and the ecological impact of human wells.

**Keywords:** poverty alleviation resettlement; ecological environment quality; remote sensing ecological index; karst ecologically fragile areas





## 1. Introduction

Climate change and human activities have an overall impact on global ecology [1,2]: on the one hand, human activities such as urban expansion and deforestation affect ecosystem degradation in most parts of the world [3,4]; on the other hand, human activities such as ecological restoration can improve ecosystems to a certain degree [5,6]. Ecological environment quality (EEQ) is the degree of suitability of the ecological environment for human survival and sustainable social–economic development within a certain space–time range [7]. Scientific monitoring and the evaluation of the impact of human activities on EEQ and temporal and spatial changes have shown important theoretical and practical significance to coordinate the relationship between human activities and the ecological environment, and to promote the sustainable development of society.

The United Nations identified poverty eradication as the primary goal of sustainable development, having invested 600 billion CNY in PAR from 2016 to 2020, involving 10 million extremely poor farmers; this is one of the flagship projects to eliminate poverty in

China [8]. The area of PAR is typical of the vicious circle of poverty and ecological environment deterioration, with a high overlap of ecological fragility and extreme poverty [9,10]. At the same time, the PAR areas also belong to the spatial poverty trap [11], and the task of eradicating poverty in situ is extremely difficult. The PAR is a better development opportunity for farmers who live below the poverty line in ecologically fragile areas, by moving them to cities and towns.

The Chinese government believes that China's PAR has positive significance for eradicating poverty and improving the ecological environment [12]. Should we consider PAR as an effective human activity to improve ecology? However, the existing research on PAR has basically focused on the social effectiveness of its mechanisms of participation and poverty reduction [13,14], with few studies being related to ecological restoration. Similar to the ecological resettlement policy, most scholars are skeptical of the ecological protection effect of the large-scale implementation of ecological resettlement in ecologically fragile areas, despite the effectiveness of ecological restoration or the welfare of relocated farmers [15,16]. However, different from ecologically fragile areas such as Tibet and Inner Mongolia, the climate conditions in the southwest karst mountainous area where PAR is mainly implemented are more suitable for plant growth (the annual average rainfall is approximately 1100–1300 mm, and the average annual temperature is approximately 16 °C), so the dominant biophysical limiting factor is not the climate, but the soil resources [17], and human interference is the main factor influencing ecological restoration [18,19]. According to peasants–land coordination theory, in regions with limited resources, human constraints and natural interference are the best choices to coordinate the relationship between peasants and land [20]. PAR reduces human disturbance to natural resources (soil resources), resulting in ecological improvement and the rapid shrinkage of inefficient agricultural production space in the relocated area, which is in line with the Environmental Kuznets Curve theory of ecological economics [21,22]. This paper aims to explore whether the PAR really contributes to the improvement of EEQ as a human activity and actually improves the living environment for human beings. If the EEQ improves the study area, can we consider it to be caused by PAR? We need to quantify the effectiveness of PAR-driven eco-environmental improvements, among which we must distinguish the influence of human activities from natural factors, which will be the focus of this paper.

Due to remote-sensing data being timely and effective, covering a wide area, and being objective and sustainable, the application of remote-sensing technology in ecological environment assessment has increasingly attracted attention from scholars [23]. Using the normalized difference vegetation index (NDVI) to assess ecological environment is the most common method [24], and most scholars use land surface temperature (LST) in evaluating the effect of the urban heat island [25]. Similarly, the wet components form tasseHed captransform (WET), and the normalized difference impervious surface index (NDISI) indicators are the most important indicators for the intuitive human perception of ecological conditions [26,27]. Compared with a single indicator, the ecological status reflected by the comprehensive indicator is more complex and diverse. RSEI, which is based on the Ecological Index (EI) from the Ministry of Ecology and Environment of China, reflects the Technical Criterion for Ecosystem Status Evaluation (HJ 192e2015). EI is authoritative and extensive in regional EEQ assessment in China [28]. Many scholars have verified that RSEI and EI are highly comparable in the ecological sense [29]. The RSEI (Remote Sensing Ecological Index) model integrates intuitive and key influence factors, including greenness, wetness, dryness, and heat. It has the advantages of real and effective evaluation data sources, objective and fair evaluation conclusions, and intuitive and visual evaluation results [30,31]. Additionally, many scholars have evaluated the EEQ improvement effect by using RSEI as a technical means in projects such as Northwest Beijing Ecological Containment Area [32], the Northern Sand-Prevention Belt [33], and the Three-North Shelter Forest Program [34]. Existing research results show that the model of the trend of normalized residuals enables distinguishing between climatic factors and ecological effects caused by human activities [35]. It is necessary to clarify the turning point

where PAR causes obvious changes in RSEI; a regression model was constructed using weather factors and human-activity factors during the time before the turning point to predict the RSEI trends during the period of PAR implementation, which was not affected by PAR, and the residual between the observed RSEI and the predicted RSEI, which was thought to be caused by PAR. Because the conclusion of the regression model is predictive, no PAR data are involved, so a correlation analysis model needs to be constructed with PAR implementation data and period RSEI variables to further explain the EEQ effect of PAR. The overall objective of this study is to evaluate the effectiveness of PAR on long-term EEQ dynamics in the study area, by the following means: (1) the analysis of the spatial and temporal evolution trend of long-series RSEI; (2) the elimination of the impact of climate factors and ecological restoration projects such as the Karst Rocky Desertification Restoration Project, to analyze the ecological contribution of the PAR; and, (3) establishing a coupling model between RSEI changes and the village-level PAR population to determine the association.

Southwest Guizhou Autonomous Prefecture, located in the Yunnan, Guizhou, Guangxi, concentrated, contiguous special-hardship area, is one of the most ecologically fragile regions in China, with few resources, a low environmental carrying capacity, a fragile ecological environment and human–land conflict [36]. From 2016 to 2019, 74,600 households with 338,600 people were relocated for the purposes of poverty alleviation, accounting for 3.38% of the total relocated population in China. This paper takes Southwest Guizhou Autonomous Prefecture as the study area, and first applies the Google Earth Engine (GEE) processing platform and cloud computing to obtain and reconstruct remote sensing data from 2000 to 2020, then applies the RSEI model to quantitatively evaluate the spatial and temporal evolution of EEQ in the study area. The ecological contribution of the PAR to the study area is quantified using the model of the trend of normalized residuals, and the association between the PAR and RSEI changes is further determined with a coupled model. Furthermore, we quantitatively reveal the changes in EEQ spatial distribution and the trend of ecological environment improvement caused by PAR, and provide theoretical support for coordinating ecological environment protection and social and economic development in ecologically fragile regions to achieve harmonious development between man and nature.

## 2. Materials and Methods

### 2.1. Study Area

The Guangxi, Yunnan, and Guizhou areas are located in the karst mountains of southwest China, with a combined population of 220 million people. They span across 0.54 million km$^2$ of carbonate rock area, which is one of the most ecologically fragile and densely populated areas in the world [37]. The Southwest Guizhou Autonomous Prefecture is located in the southwestern part of Guizhou Province between 104°35′–106°32′ E and 24°38′–26°11′ N. It has eight counties under its jurisdiction and a land area of 16,800 km$^2$. The area belongs to the subtropical humid monsoon climate and has the most widely distributed carbonate rock layer containing magnesium in the Triassic marine. The karst area in the region is spread across 10,200 km$^2$, accounting for 60.28% of the total land area. Dominant ecological problems in Southwest Guizhou Autonomous Prefecture are stone desertification and soil erosion. The potential stone desertification is spread across 0.21 million km$^2$, where the known stone desertification area is 0.50 million km$^2$, accounting for 42.51% of the land (Figure 1), making it one of the most severely ecologically compromised areas of China [38]. With a rural-poor population of 432,300 in 2015 and a poverty incidence rate of 13.75%, the problem of poverty is relatively prominent. Southwest Guizhou Autonomous Prefecture is a region with high overlap between ecological fragility and extreme poverty.

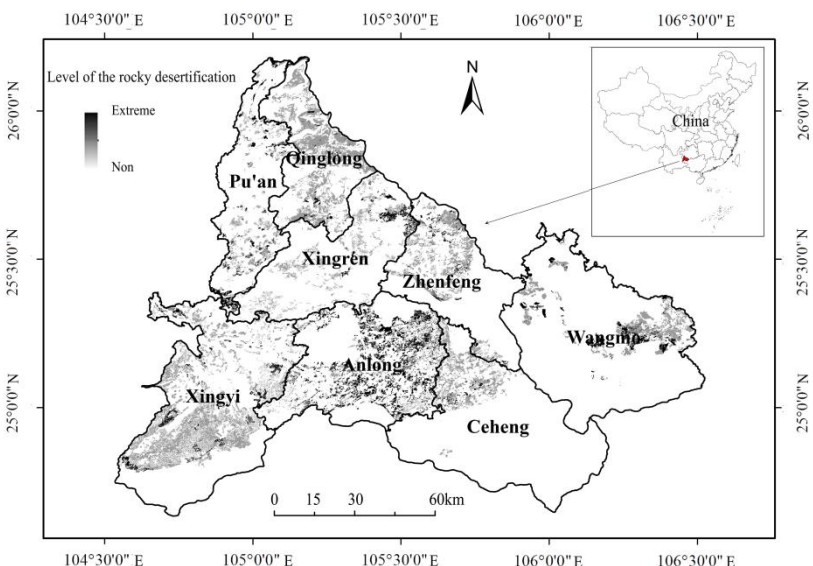

**Figure 1.** Distribution of degree of stone desertification in the study area.

PAR is the flagship project of the Chinese government's poverty alleviation project. For people living in areas where local resources cannot effectively carry them out of poverty, they relocate to urban areas with better education, medical care, transportation, communication, employment and other improved conditions. Here, they enjoy the favorable development resources of the city, which help them to become free of poverty. The Chinese government concluded that PAR has 9 key achievements, including improved living conditions, broader employment prospects, and the relief of ecological environment pressure [12]. PAR in Southwest Guizhou Autonomous Prefecture involved 1222 villages, 74,600 households and 338,600 people from 2016 to 2019, all of whom were farmers living below the poverty line. The areas from which people were relocated, from low-resource areas to high-resource areas, were Xingyi, Xingren, Anlong, Pu'an, Zhenfeng, Wangmo, Qinglong, and Ceheng, which are concentrated in the north and southeast (Figure 2). The relocated farmers were resettled in 65 resettlement sites in cities and towns, accounting for 99.74% of the total relocated population (the other 0.26% were resettled in centralized rural areas, being relatively scattered), and 26 resettlement sites, with more than 5000 people each, resettled a total of 242,567 people, accounting for 71.66% of the total resettlement. Southwest Guizhou Autonomous Prefecture is dominated by centralized resettlement in cities (Figure 2).

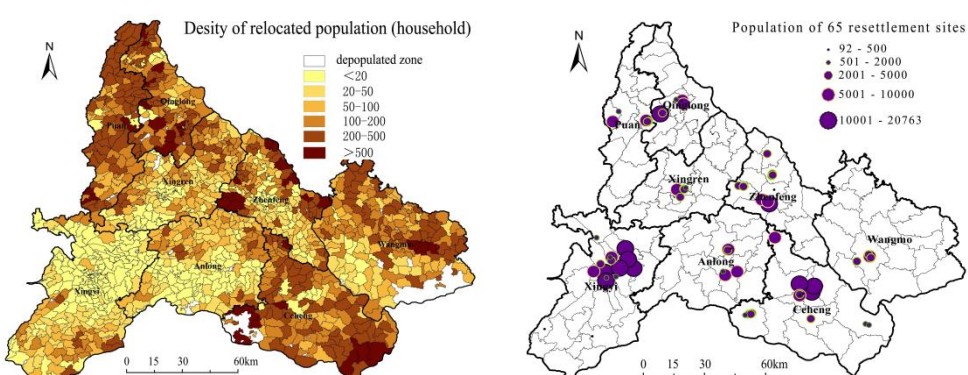

**Figure 2.** Density of relocated household and distribution of 65 resettlement sites of PAR.

*2.2. Data Resources and Pre-Processing*

The remote-sensing data were mainly obtained from Landsat in the GEE platform database, including Landsat 8 (OLI) data for 2013–2020 and Landsat 5 (TM) data for

2000–2012, with a spatial resolution of 30m and a temporal resolution of 16 d. In addition, considering the cloudy climatic attributes of Guizhou Province, effective remote-sensing images of low cloud cover could not be collected in the summer. Therefore, the study screened automatic synthetic Landsat images from April to October for the target years. The GEE official programming algorithm was then used to pre-process data, and to complete geometric correction, radiation correction and atmospheric correction. We achieved cloud-removal processing through the cloud-mask algorithm. In addition, the updated mask was implemented by code to avoid the effect of water area on the load distribution of the principal components. After using the GEE programming calculus to obtain standardized remote-sensing data on the study area from April to October, vector data on the administrative regions at all levels in Southwest Guizhou Autonomous Prefecture State were obtained from the Resource and Environment Science Data Center of the Chinese Academy of Sciences (http://www.resdc.cn, 15 December 2021), the Grain to Green Project, the Karst Rocky Desertification Restoration Project from the Master Plan of National Forestry and Grassland Administration (http://www.forestry.gov.cn, 12 March 2022), and the National Development and Reform Commission (http://www.rdrc.gov.cn, 12 March 2022). The meteorological data were obtained from the China Meteorological Administration Network (http://www.cma.gov.cn, 12 March 2022), and population data for the PAR were obtained from the Ecological Migration Bureau of the Guizhou Province (10 September 2021).

*2.3. Methodology*

The RSEI model first proposed by Xu et al., was related to four indicators—greenness, wetness, heat, and dryness—which can be visually determined and are widely used to understand the quality of the ecological environment.

The entire data calculation process was based on the GEE online platform, ArcGIS software, and ENVI software. GEE is the most critical platform for original data acquisition, preprocessing, and RSEI calculation. The GEE operation process was as follows: determine the scope and timeliness; clarify the image type; function cloud mask; function remove cloud; calibrated radiance; function normalization; use unit scale to normalize the pixel values; calculate the NDWI; calculate the NDVI, WET, LST, and NDISI (formula 1–9); collection merge; visualization; map; function PCA model; eigenvalue, eigenvector; return result; normalize the RSEI; and export image to drive (see attachment for original data). ArcGIS and ENVI further process the original data in order to meet the research needs.

(1) Calculation of component indexes. Among the four indexes, the greenness index reflects the regional vegetation coverage, and the normalized difference vegetation index (NDVI) is closely related to the leaf area index and vegetation coverage. The wetness indicators (WET) reflect the moist conditions of the regional surface objects and are expressed as the wet components of a tasseHed captransform of the surface vegetation, soil, etc. Here, the formulae used to calculate EM and OLI were somewhat different. The dryness index reflected the surface drying condition; the soil index and the index-based built-up index IBI were expressed as the normalized difference impervious surface index (NDISI). Heat indicators reflected the surface temperature conditions and were expressed by the land surface temperature (LST). Based on the existing research results, the four indicators were calculated as follows:

$$NDVI = \frac{B_{NIR} - B_{red}}{B_{NIR} + B_{red}} \tag{1}$$

$$WET_{TM} = 0.0315B_{blue} + 0.2021B_{green} + 0.3102B_{red} + 0.1594B_{NIR} - 0.6806B_{SWIR1} - 0.6109B_{SWIR2} \tag{2}$$

$$WET_{OLI} = 0.1511B_{blue} + 0.1972B_{green} + 0.3283B_{red} + 0.3407B_{NIR} - 0.7117B_{SWIR1} - 0.4559B_{SWIR2} \tag{3}$$

$$NDBSI = \frac{SI + IBI}{2} \tag{4}$$

$$SI = \frac{(B_{SWIR1} + B_{red}) - (B_{NIR} + B_{blue})}{(B_{SWIR1} + B_{red}) + (B_{NIR} + B_{blue})} \tag{5}$$

$$IBI = \left\{ 2 \times B_{SWIR1}/(B_{SWIR1} + B_{NIR}) - [B_{NIR}/(B_{NIR} + B_{red}) + B_{green}/\{(B_{green} + B_{WIR1})\}]\right\}/2 \times B_{SWIR1}/(B_{SWIR1} + B_{NIR})$$
$$+[B_{NIR}/(B_{NIR} + B_{red}) + B_{green}/(B_{green} + B_{WIR1})]\}$$
(6)

$$LST = \frac{K_2}{\ln(K_1/B(T_S) + 1)}$$
(7)

$$B(T_S) = \frac{L_{10} - L_{up} - \tau_{10}(1 - \varepsilon_{10})L_{down}}{\tau_{10}\varepsilon_{10}}$$
(8)

$$L_{10} = \tau_{10}[\varepsilon_{10}B(T_S) + (1 - \varepsilon_{10})L_{down}] + L_{up}$$
(9)

Here, *NDVI* indicates greenness, *WET* indicates humidity, *NDISI* indicates dryness, and *LST* indicates heat. The specific meaning of each variable in the equation is referred to in reference [29].

(2) Calculation of RSEI. Based on the results of the four indicators, NDVI, WET, NDSI, and LST, they were first normalized by using positive normalization to standardize their values. The initial RSEI values were calculated using NEVI software by principal component analysis of the standardized data from the four indicators for PC1. Indicator weights that were not dictated by humans were considered in the calculation of the initial RSEI value. The final RSEI value was obtained using the forward normalization process, where the value was between 0 and 1. The higher the numerical value, the better the EEQ.

$$NI = \frac{I - I_{min}}{I_{max} - I_{min}}$$
(10)

$$RSEI_0 = PC1[f(NDVI, WET, NDBSI, LST)]$$
(11)

$$RSEI = \frac{RSEI_0 - RSEI_{0-min}}{RSEI_{0-max} - RSEI_{0-min}}$$
(12)

In the above formula, *NI* denotes the standard index value after processing; *I* is the index value; and $I_{max}$ and $I_{min}$ are the maximum and minimum values of the index, respectively. *RSEI* denotes the final remote sensing ecological index; $RSEI_0$ denotes the primary remote sensing ecological index. $RSEI_{0\text{-}max}$ and $RSEI_{0\text{-}min}$ are the maximum and minimum values of the primary remote-sensing ecological index in the current period, respectively. *PC1* denotes the first principal component.

(3) Residual trends method. RSEI changes are influenced by climatic conditions and human activities. We used the residual trends method to calculate the extent to which PAR contributes to RSEI. Based on the mean value of RSEI from 2000 to 2020, the turning point of RSEI was determined according to linear trend analysis. A multiple linear regression analysis model was established with RSEI (dependent variable) and four factors (independent variables). These were the natural factors—mean annual temperature (MAT) and annual total precipitation (ATP)—and the human activity factors, outlined in the Grain to Green Project and the Karst Rocky Desertification Restoration Project, that may affect EEQ changes during the reference period (before the turning point).

$$V_{D,T}(RSEI) = a \times V_{I,T}(MAT) + b \times V_{I,T}(ATP) + c \times V_{I,T}(\textit{funds of the Grain to Green Project}) +$$
$$d \times V_{I,T}(\textit{funds of the Karst Rocky Desertification Restoration Project})$$
(13)

where $V_{D,T}$ denotes the dependent variables at a specific time, $V_{I,T}$ denotes the independent variables at a specific time, *a* and *b* are multivariate regression standardization coefficients of climate conditions, while *c* and *d* are multivariate regression standardization coefficients of human activities. Formula (13) was used to calculate the predicted RSEI value during project implementation (only affected by four factors) with a 95% confidence interval; the residual between the observed RSEI mean value and the predicted RSEI mean value was calculated. If the residual value was positive, PAR was assumed to have a positive impact on EEQ over time. If the residual was negative, PAR was assumed to harm the local ecology.

(4) Analysis of coupling coordination degree. To test the degree of spatial and temporal correlation between PAR and EEQ, this study used the coupling coordination degree to analyze and describe the spatial characteristics between PAR and ecological environment variables. The range of values was [0, 1], where a larger coupling coordination degree indicated a stronger correlation between the two variables [39].

$$C = \sqrt{\frac{U_1 \times U_2}{[(U_1 + U_2)/2]^2}} \tag{14}$$

$$D = \sqrt{C \times T}; \; T = \alpha \times U_1 + \beta \times U_2 \tag{15}$$

Using formula $C$ in particular, the system coupling degree was calculated. Here, $U_1$ and $U_2$ are the RSEI variables, and the data were normalized for the size (number of people) of the relocated population. $T$ is the comprehensive evaluation index of $U_1$ and $U_2$; $\alpha$ and $\beta$ are the undetermined coefficients, where $\alpha = \beta = 0.5$; $D$ is the coupling coordination degree. The coupling coordination degree is based on SPSS software, which normalizes and standardizes the analysis data. The conclusion of calculations is between 0 and 1, which the software automatically divides into 10 levels. According to the trend in the level distribution and the existing method of dividing the research results, the coupling coordination level is divided into 4 levels: extreme detuning (0–0.3), general detuning (0.3–0.5), general coordination (0.5–0.7), and extreme coordination (0.7–1) [40].

## 3. Results

### 3.1. Ecological Environment Quality, Spatial and Temporal Evolution, and Driving Forces

3.1.1. Evolution Trend Analysis

According to the statistical analysis of the average EEQ value from 2000 to 2020 (Figure 3), the overall EEQ of the study area showed an increasing trend. The mean value of RSEI increased from 0.5329 in 2000 to 0.6363 in 2020, i.e., an increase of 0.1034 or 19.40% in 20 years. The overall trend of EEQ in the research area is still improving steadily. The overall trend of RSEI from 2000 to 2011 was relatively stable, although there were some fluctuations. From 2011 to 2016, the RSEI began to improve, although, again, there were still some fluctuations. The overall trend showed a steady improvement. From 2016 to 2020, RSEI in the study area began to improve significantly and remained stable. According to the evolution trend, the turning point of RSEI in the study area was determined to be 2016. In combination with the human activity in the area, we focused on analyzing the spatial and temporal distribution of RSEI to achieve the research objectives of this paper. The Grain to Green Project began in 2000; the Karst Rocky Desertification Restoration Project was implemented in 2008; and the PAR was implemented in 2016. Considering the time interval of the study period and that the effect of ecological restoration projects is somewhat delayed, we focused on data of the spatial and temporal distribution of RSEI for the years 2000, 2005, 2010, 2015, and 2020.

3.1.2. Spatial–Temporal Evolution Analysis of EEQ

For convenient comparison, the average RSEI value was divided into five grades according to the average value: poor (0–0.2), relatively poor (0.2–0.4), moderate (0.4–0.6), good (0.6–0.8), and excellent (0.8–1.0) [41]. The spatial distribution of RSEI values in the study area was characterized by higher EEQ grades in the southeastern part of Ceheng and Wangmo, and all regions were optimized after 2010 to some extent.

During 2000–2010, the EEQ values of the study areas were generally similar, i.e., excellent and good grades accounted for ~44% of the subject areas, particularly in the southeastern areas of Ceheng and Wangmo (Figure 4).

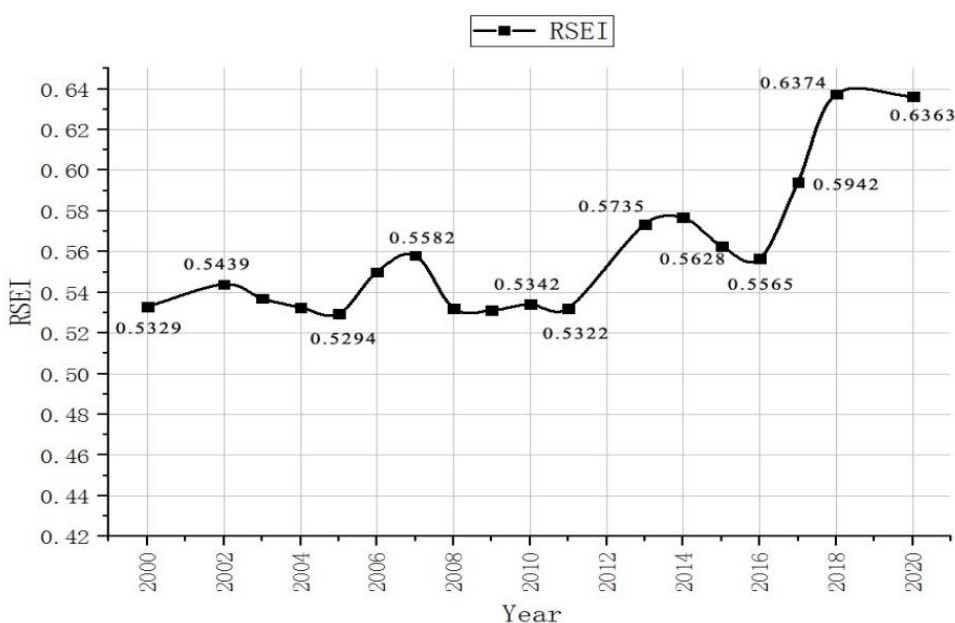

**Figure 3.** Mean RSEI value during 2000–2020.

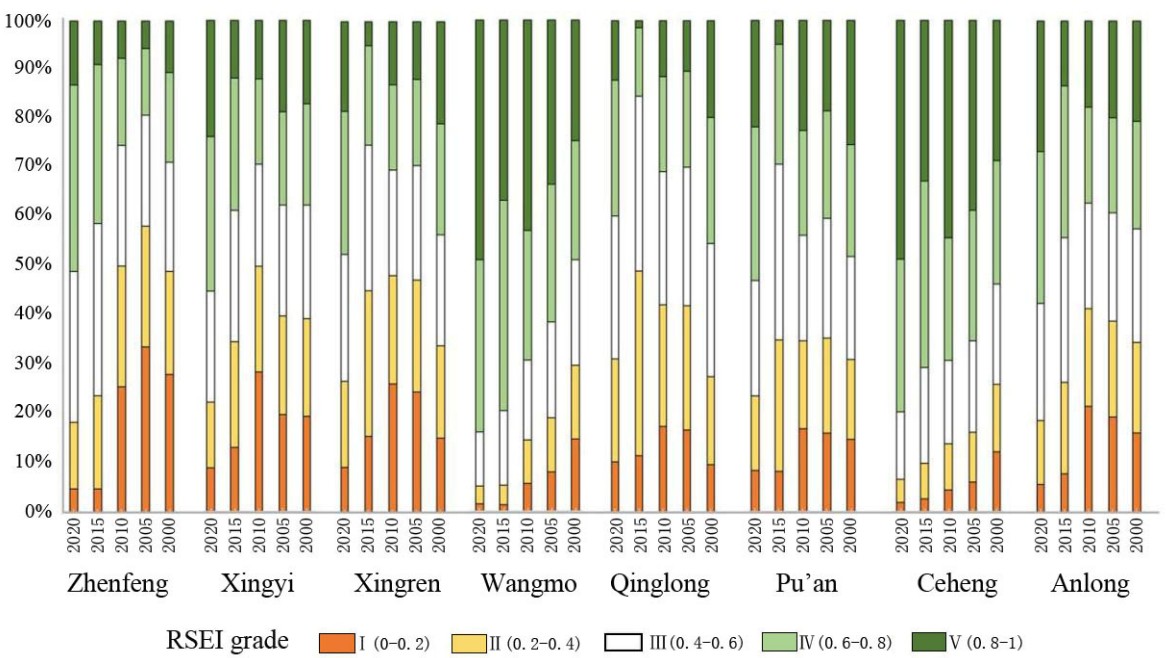

**Figure 4.** Eco-environment quality dynamics by classified RSEI of the counties from 2000 to 2020.

During the period 2010–2015, the proportion of EEQ attributable to poor and relatively poor grades decreased to 26.26%, which originated from areas mainly concentrated in north-central Xingren, south-central Pu'an, north-central Xinyi, and other regions. The proportion of excellent and good grades rose to 48.46%, which was attributable to the areas that were concentrated in the southeastern area of Ceheng and Wangmo. During this period, the EEQ grade remained the same in an area accounting for 42.48% of the total study area. The area that saw an EEQ reduction accounted for 24.80% of the study area. This area with reduced EEQ was mainly distributed in Wangmu, Ceheng, Pu'an, and Qinglong. In this period, the dominant EEQ grades were excellent and good (Figures 4 and A1, Table 1).

**Table 1.** The transition matrix of EEQ levels during 2010–2015.

| EEQ Level | I | II | III | IV | V |
|---|---|---|---|---|---|
| I | 830.13 | 1143.01 | 644.35 | 209.61 | 40.58 |
| II | 261.50 | 1038.92 | 1088.78 | 408.93 | 80.93 |
| III | 93.70 | 600.43 | 1471.60 | 1013.50 | 208.03 |
| IV | 32.88 | 185.00 | 892.54 | 1777.49 | 576.91 |
| V | 11.96 | 51.58 | 294.53 | 1679.01 | 1910.32 |

During the period from 2015 to 2020, the proportion of areas with an EEQ grade of poor or relatively poor decreased to 17.64%, and the distribution was mainly concentrated in the key areas of urban development. The proportion of areas with excellent and good grades further increased to 63.35%, with the most concentrated contiguous areas in Ceheng, Wangmo, and southeast of Zhenfeng. The EEQ improvement regions were most concentrated in Qinglong and Pu'an, which were the areas with the most fragile ecological environment areas and the most prominent peasants–land conflicts. The area of reduced EEQ was mainly concentrated in the urban development areas, such as Xingyi, where the population was further concentrated, and socio-economic development was more important (Figures 4 and A1, Table 2).

**Table 2.** The transition matrix of EEQ levels during 2015–2020.

| EEQ Level | I | II | III | IV | V |
|---|---|---|---|---|---|
| I | 395.94 | 407.33 | 303.77 | 98.76 | 22.56 |
| II | 337.98 | 830.39 | 1175.63 | 566.99 | 102.63 |
| III | 137.48 | 475.53 | 1383.03 | 1892.35 | 489.99 |
| IV | 41.54 | 127.90 | 497.46 | 2215.79 | 2177.92 |
| V | 10.00 | 29.22 | 99.47 | 541.48 | 2098.71 |

### 3.1.3. RSEI Result Test

The Pearson correlation coefficient was used to test the RSEI results. The results are presented in Table 3, where the average correlation of RSEI and the four indicators reached a maximum of 0.953, 0.964, −0.739 and −0.945, respectively. The correlation between RSEI and NDVI, WET, NDISI was significant at the 0.01 level, indicating strong significance, and the correlation with LST was significant at the 0.1 level, indicating general significance.

**Table 3.** Correlation matrix of indexes during 2000–2020.

| Indicator | NDVI | WET | LST | NDISI | RSEI |
|---|---|---|---|---|---|
| NDVI | 1 | 0.969 | −0.869 | −0.814 | 0.953 |
| WET | 0.969 | 1 | −0.804 | −0.831 | 0.964 |
| LST | −0.869 | −0.804 | 1 | 0.56 | −0.739 |
| NDISI | −0.814 | −0.831 | 0.56 | 1 | −0.945 |
| RSEI | 0.953 | 0.964 | −0.739 | −0.945 | 1 |
| Sig. | 0.012 | 0.008 | 0.154 | 0.015 | |

### 3.2. Drivers of Change in EEQ

Unlike other karst areas in the world, where the population density is low, the karst mountains in southwest China are populous and ecologically fragile. There is sparse coordination between peasant–land conflicts and high ecological pressure [42]. Various ecological restoration projects have been promoted since 2000, such as the Grain to Green Project (from 2000) and the Karst Rocky Desertification Restoration Project (from 2008) [43]. According to the mean values of the four indicators presented in Table 4, the overall indexes were relatively stable, and the average RSEI values were also relatively stable.

**Table 4.** Mean indicators of RSEI from 2000 to 2020.

|      | NDVI   | WET    | LST    | NDISI  | RSEI   |
|------|--------|--------|--------|--------|--------|
| 2000 | 0.5555 | 0.5286 | 0.5627 | 0.5073 | 0.5329 |
| 2005 | 0.5430 | 0.5266 | 0.5008 | 0.5046 | 0.5294 |
| 2010 | 0.5398 | 0.5411 | 0.5518 | 0.5120 | 0.5342 |
| 2015 | 0.6689 | 0.5642 | 0.4279 | 0.5095 | 0.5628 |
| 2020 | 0.7511 | 0.5952 | 0.4307 | 0.4217 | 0.6363 |

During 2010–2015, the mean value of NDVI increased by 23.92%, and the WET mean value also increased significantly. The effectiveness of the ecological restoration project is also highlighted during this period, having contributed to a significant decrease in the mean value of LST. The ecological restoration project also contributed to an increase in the mean value of RSEI in the study area (Table 4).

From 2015 to 2020, PAR was promoted and completed. During this period, NDISI was reduced significantly, by 17.24%. Relocation to alleviate poverty caused many rural people to move to towns and cities, while their original home base was reclaimed and re-greened. In this case, the impact of human activities on rural areas was significantly reduced, which directly contributed to a significant reduction in the dryness index. The NDVI was increased, along with the WET, which means that the conflict between humans and land was fundamentally relaxed, and the effectiveness of various ecological restoration projects was maintained and further improved. PAR was a key factor in the improvement of RSEI in the study area (Table 4).

### 3.3. The Contribution of PAR to EEQ Changes

A multiple linear regression analysis model was used to calculate the residual trends in PAR implementation, whereby the turning point of RSEI change caused by PAR was determined to occur in 2016. The variation in RSEI was influenced by natural factors, including MAT and ATP, as well as human activities, including the Grain to Green Project and the Karst Rocky Desertification Restoration Project. We took these four factors as the independent variables, i.e., the influencing factors, and RSEI was adopted as the dependent variable, i.e., the resulting factor. Standardized coefficients of the regression models were analyzed by the SPSS with 95% confidence (Table A1). The results show that the observed cumulative probability and the predicted cumulative probability were normally distributed in the linear regression analysis model (Figure 5). Furthermore, the standardized residuals were randomly distributed without outliers (Figure 6), and the regression model significance was 0.038 at a significance level of 95%. Therefore, the multiple linear regression equation was verified to be stable. The analysis results show that the RSEI variable could mathematically be represented as Variable (RSEI) = $0.273 \times$ Variable(MAT) + $0.285 \times$ Variable(ATP) $- 0.144 \times$ Variable (funds of the Grain to Green Project) + $0.520 \times$ Variable (funds of the Karst Rocky Desertification Restoration Project). This equation predicted the mean RSEI values in 2017, 2018, and 2020 to be 0.5676, 0.5660, and 0.5701, respectively, under the influence of the four factors. The observed RSEI values in 2017, 2018, and 2020 were 0.5942, 0.6374 and 0.6363, respectively. The residual difference between the observed and predicted mean values of RSEI was likely due to the PAR-driven RSEI improvement. The improvements in RSEI caused by PAR were 0.0266, 0.0715, and 0.0662 in 2017, 2018, and 2020, respectively, according to the actual increase of 0.0266, 0.0715, and 0.0662. Compared with the turning point of 2016, the PAR contributed to the RSEI growth contribution rate of 70.56%, 88.38%, and 82.96% in the three years, respectively. Collectively, after 2016, the RSEI values increased significantly, and from there on, the average RSEI reached good levels and remained relatively stable.

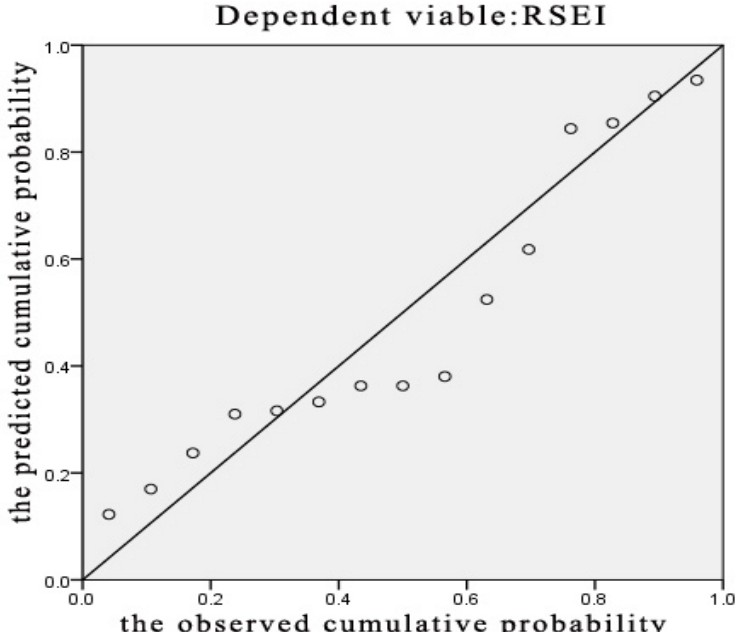

**Figure 5.** P-P plot of regression standardized residuals.

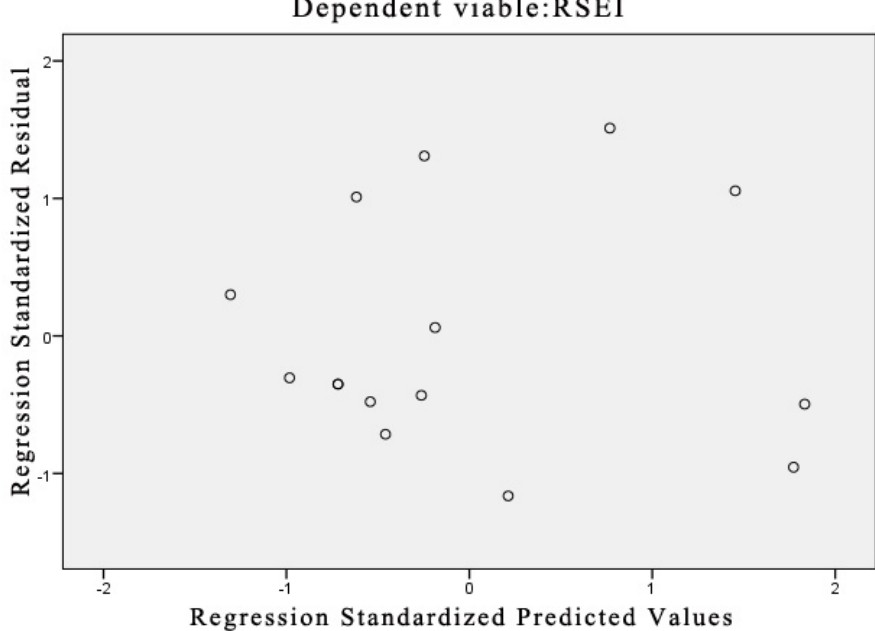

**Figure 6.** Scatter diagram.

*3.4. Correlation between PAR and EEQ Changes*

The correlation between relocation and EEQ changes was analyzed by taking the village area as the basic unit. A map showing the change in RSEI during 2015–2020 is shown in Figure 7.

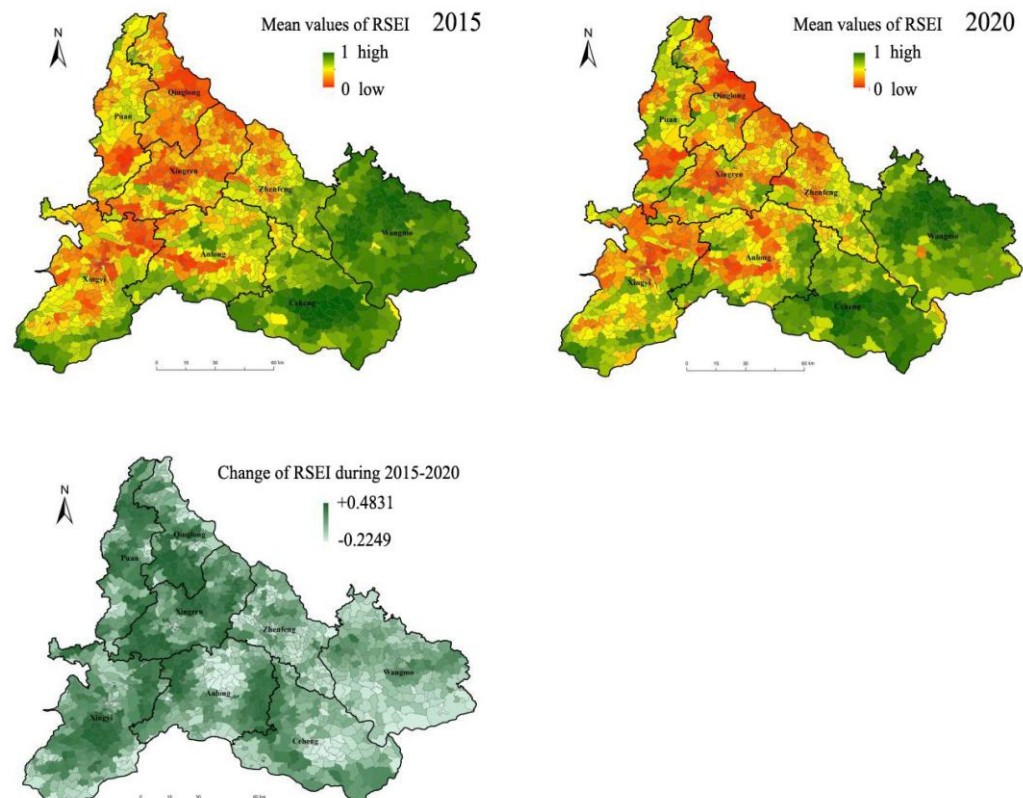

**Figure 7.** Change map of RSEI during 2015–2020.

There were 1330 administrative village units in the study area. The number of villages in the area with reduced RSEI was 205, accounting for 15.41% of the total. A total of 1125 villages exhibited improved RSEI. These villages were mostly located in the central and northern regions of Pu'an and Qinglong. The original EEQ of these regions was relatively low, and the effect of the upgrade in terms of RESI was obvious. The relocation of the five counties on the northeast side of the study area of Southwest Guizhou Autonomous Prefecture, including the counties of Ceheng, Qinglong, Wangmo, Zhenfeng, and Pu'an, was the most concentrated, whereby 74,600 households were relocated, accounting for 82.47% of the total (Figure 2).

Figure 8 shows the spatial distribution of the coupling coordination between the relocation population density and RSEI variables in 1222 administrative villages in eight counties. There were 247 villages with extreme coordination, accounting for 25.33% of the area, mainly concentrated in Ceheng, Wangmo, Qinglong, and Pu'an, which are the most densely relocated areas. There were 411 general coordination villages, accounting for 33.90% of the area, most of which were located in the northeast and southeast of the study area. There were 403 general detuning villages, accounting for 29.99% of the area, and 161 extreme detuning villages, accounting for 7.63% of the total villages. These villages were predominantly located in economically developed areas, such as Xinyi. After the spatial analysis of the coupling coordination between relocation population density and RSEI variables, a significant coupling coordination relationship was found between EEQ enhancement and PAR in the study area.

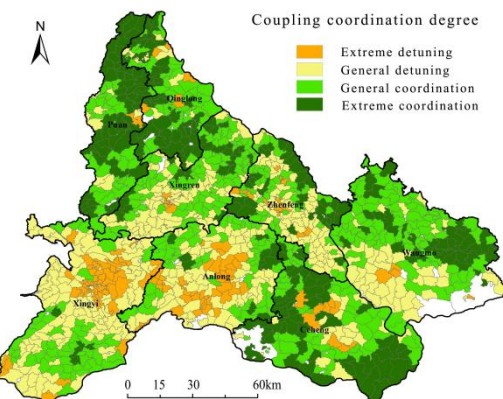

**Figure 8.** Coupling coordination degree of density of relocated population, and change in RSEI.

*3.5. Forecast of Future RSEI*

The RSEI of the study area was predicted after the implementation of the PAR to analyze its sustained impact on regional EEQ. The ARIMA model starts from the time-series itself and forecasts future data based on past behavioral data [44]. The expert modeler was used for prediction, where the autocorrelation and partial correlation coefficients of the model were all within the confidence zone of 95%. After prediction, the mean value of RSEI in the study area was predicted to improve to 0.7355 in 2025, and to 0.8603 in 2030. Continuous improvement and optimization were maintained from thereon. The spatial distribution effects are shown in Figure 9.

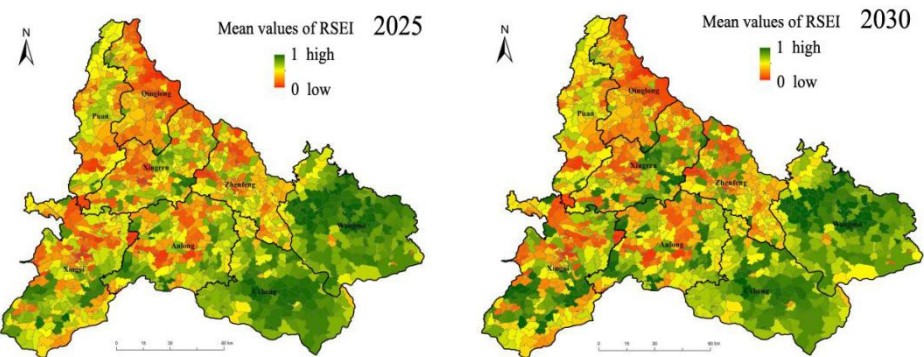

**Figure 9.** Forecast of the mean values of RSEI.

**4. Discussion**

The results show that not all areas with a large number of relocated people have significantly increased RSEI. In northern Pu'an, Qinglong, and other severely rocky desertification areas (Figure 1) where the population density is relatively large (Figure A2) and the contradiction between peasants and land is prominent, the RSEI increase is more obvious in areas with a large number of people involved in PAR (Figure 7). However, in non-rocky desertification areas such as Ceheng and Wangmo in the southeast, where the population density is relatively small, the RSEI does not have an obvious increase, although the number of people relocated is relatively large. This shows that the implementation of PAR is more effective in areas with relatively poor ecology and a prominent contradiction between peasants and land, while in areas where the ecology is better, the contradiction between peasants and land is not prominent, and the ecological effect is not obvious (Table 5).

**Table 5.** Summary table of positive and negative effects of PAR.

| Indicators | | Positive Effects | Negative Effects | Remark |
|---|---|---|---|---|
| | Society | Improved infrastructure, education, employment and health care, etc. | Increased cost of living, changed way of life, blocked cultural inheritance, etc. | Will not be discussed |
| | Sustainability | The remaining farmers will have more production resources, and will obtain more employment opportunities. | Traditional culture is destroyed, increasing living costs, and increased burden on the government. | |
| | Spatial features | The RSEI of the entire study area significantly improved. | On the southeast side where the contradiction between peasants and land is not prominent, the RSEI improvement is not obvious. | |
| | Indicator structure | Increase in NDVI and decrease in NDISI. | The increase in WET is small, and the negative indicator Lst increases. | Unbalanced RSEI promotion |
| | Biodiversity | Human activity is reduced in abandoned areas, which is beneficial to biodiversity. | After PAR, the planting structure will tend to be single-species, thus affecting the biodiversity. | |
| Ecology | Human well-being | The overall EEQ of the study area improved, and the entire area became greener. | There is a certain distance of EEQ between the ecologically improved area and the gathering area of the relocated farmers. | |
| | Applicability | PAR can promote ecological improvement in karst ecologically fragile areas where the contradiction between peasants and land is prominent. | PAR may not be effective in ecological restoration in areas where climate is the main limiting factor (such as water resources in semi-arid regions). | |
| | Other aspects | The relocation area can make more convenient and intensive use of land, and more effectively promote the implementation of various ecological restoration projects. | May possibly cause single-species forest and the waste of water resources (in the context of global warming, extreme dry weather has already occurred in 2011). | |

In terms of the effectiveness of EEQ, the improvement in RSEI in the study area contributed by PAR is mainly due to the increase in NDVI and the decrease in NDISI. After relocation, the large-scale management of forestland will become a trend, with farmers moving away from contracted land, and farmland being transformed to forest, which contributes to the substantial increase in NDVI. Whether the substantial increase in NDVI will cause single-species forest and excessive water consumption needs further consideration [45]. The increase in WET is significantly smaller than the increase in NDVI, which is further verified. The RSEI of the study area is indeed significantly improved, while the RSEI of the urban areas to which farmers are moved is reduced. Does such an increase in RSEI really improve the living environment for local humans? During the implementation of PAR, the overall increase in RSEI in the study area was partially reduced (with a decrease in the more concentrated areas of population) and the comprehensive indicators showed improvement. Whether such improvement is conducive to sustainable social development is something our follow-up study needs to consider deeply (Table 5).

Biodiversity has an important impact on ecosystem services, and the impact of human activities on biodiversity is the focus of various scholars and organizations such as the United Nations Environment Programme (UNEP) [46,47]. After PAR, the number of land

managers decreased and land-use patterns changed, which shifted the trend towards large-scale land management. For the purpose of easy management and economic efficiency, the planting structure of the land tends to be homogeneous after large-scale management, which leads to an impact on biodiversity and the weakening of symbiosis among plants, thus further affecting the ecosystem service function [48]. In contrast, some areas become similar to ecological reserves, where the disturbance of human activities is reduced, and biological succession proceeds in an orderly manner, which is beneficial to biodiversity. There are many influencing factors and mechanisms of internal change of biodiversity, such as human disturbance, environmental factors, management methods, etc. The impact of PAR on biodiversity needs to be further studied in future work [49,50] (Table 5).

The global trend in terms of helping rural areas is to promote traditional and sustainable farming with nature-friendly measures, rather than relocating the rural population to cities [51]. PAR is the transfer of farmers from rural to urban areas, which destroys traditional rural culture, increases their cost of living and changes their way of life. The fact that farmers move to cities to take up jobs they are not good at, and that employment training is not sufficient to help low-ability farmers relocate successfully, also greatly increases the burden on the government, in addition to the fact that the minimum cost of living for relocated households is approximately 70% higher in cities than in rural areas. PAR poses a great challenge to the sustainability of farmers [17]. The owners of the relocated rural areas are not those executing PAR, but some of the poorest farmers. The most crucial role of PAR is to alleviate the contradiction between peasants and land. The remaining farmers will have more production resources (such as renting the land of the relocated farmers), and may also obtain more employment opportunities (there will be some vacancies for forest-protection work and road-cleaning work after the relocation) and development opportunities. Relocating farmers to live together in a concentrated area will also facilitate the establishment of infrastructure such as medical care, education, training, factories, etc., to a certain extent, and it is easier to accept new knowledge, which will help farmers to achieve sustainability (Table 5).

The GEE platform was used to address the problem of the difficult acquisition of effective Landsat images under cloudy and foggy weather in Guizhou. With its powerful processing capabilities, GEE provided a foundation for the accurate analysis of the temporal and spatial patterns and evolution of EEQ. The RSEI model allows the objective analysis of EEQ. The effects of PAR and climatic factors and other ecological restoration projects on the EEQ changes could be distinguished scientifically using the residual trends. The results could then be used to establish a high and low series of coupling coordination levels between the number of relocated populations and EEQ variables, and to clarify the effective degree of correlation between PAR and EEQ enhancement. Furthermore, based on the ARIMA model, a prediction of future ecological trends concluded that the relocation of impoverished residents has a significant and sustainable driving effect on the promotion of regional EEQ. However, the RSEI model needs to use a water body mask to ensure the normalization accuracy of RSEI [30]. The Southwest Guizhou Autonomous Prefecture belongs to the ecological protection barrier in the upper reaches of the Pearl River, and 1.86% of the water area has important ecological services that unfortunately cannot be effectively reflected [41]. Due to the influence of cloudy weather in Guizhou, it is difficult to collect remote-sensing data in the same period for different years. The base data used in this paper were Landsat data from April to October of the target years. The time span was large, and there may be bias in the measurement of RSEI each year. There are other shortcomings, which provide a basis for the key research directions of the future.

PAR in karst mountains can improve local EEQ to a certain extent, but there have been some controversies. In later studies, researchers further broadened their research scope to investigate the effect of PAR on the regional carbon neutral effect [52] from the direction of carbon neutrality [53].

## 5. Conclusions

The residual trends method eliminated the confounding effects of other influencing factors and clarified the contribution of PAR to RSEI growth. The PAR made a contribution to RSEI improvement with a 13.07% increase in the study area during 2015–2020; this RESI was significantly higher than the increase noted during 2000 to 2015. The EEQ of the most ecologically vulnerable areas, such as Qinglong and Pu'an, was most significantly improved after the PAR. However, the EEQ of economically developed areas, such as Xingyi, showed a decreasing trend.

After the implementation of PAR, a large number of rural houses were dismantled and the land reclaimed as green regions. Additionally, many rural construction projects did not advance, contributing to an obvious decrease in the mean value of NDISI (to 17.24%). Furthermore, the core population of farmers moved from cultivated land to urban employment, leaving much farmland abandoned or available for planting trees, contributing to a sharp increase in the NDVI (to 12.28%). The residual trends model predicted that the mean values of RSEI in 2017, 2018, and 2020 were 0.5676, 0.5660, and 0.5701, respectively, under the influence of a multitude of factors, except for the PAR. The observed mean values of RSEI were 0.5942, 0.6374, and 0.6363, respectively. The improvement in RSEI caused by PAR was 0.0266, 0.0715, and 0.0662 in 2017, 2018, and 2020, respectively.

The spatial distribution of the coupling coordination between the relocation population density and RSEI variables showed that there was a significant positive correlation between the increase in RSEI and the relocation population density. The larger the relocated population, the greater and more significant the increase in RSEI in the region. This also led to a higher corresponding level of coupling coordination. As measured by the ARIMA prediction model, the EEQ of the study area, with many people undergoing PAR and in a cluster shape, will continue to evolve for the better. Furthermore, the EEQ of the developed urban areas represented by Xingyi will continue to decrease.

**Author Contributions:** Conceptualization, Z.Z.; data curation of PAR, W.L.; data curation of Landsat, L.W. and X.Z.; writing—original draft preparation, Q.F.; writing—review and editing, Z.Z. and Q.F.; visualization, Q.F.; supervision, C.Z.; project administration, L.Z. All authors have read and agreed to the published version of the manuscript.

**Funding:** This research was funded by the National Natural Science Foundation of China (41661088), jointly funded by the Program in Guizhou Planning of Philosophy and Social Science (21GZZD39) and the High-level Innovative Talents Training Program in Guizhou Province (2016-5674).

**Institutional Review Board Statement:** Not applicable.

**Informed Consent Statement:** Not applicable.

**Data Availability Statement:** The data presented in this study are available on request from the corresponding author. Some of the data are not publicly available due to privacy constraints.

**Conflicts of Interest:** The authors declare no conflict of interest.

# Appendix A

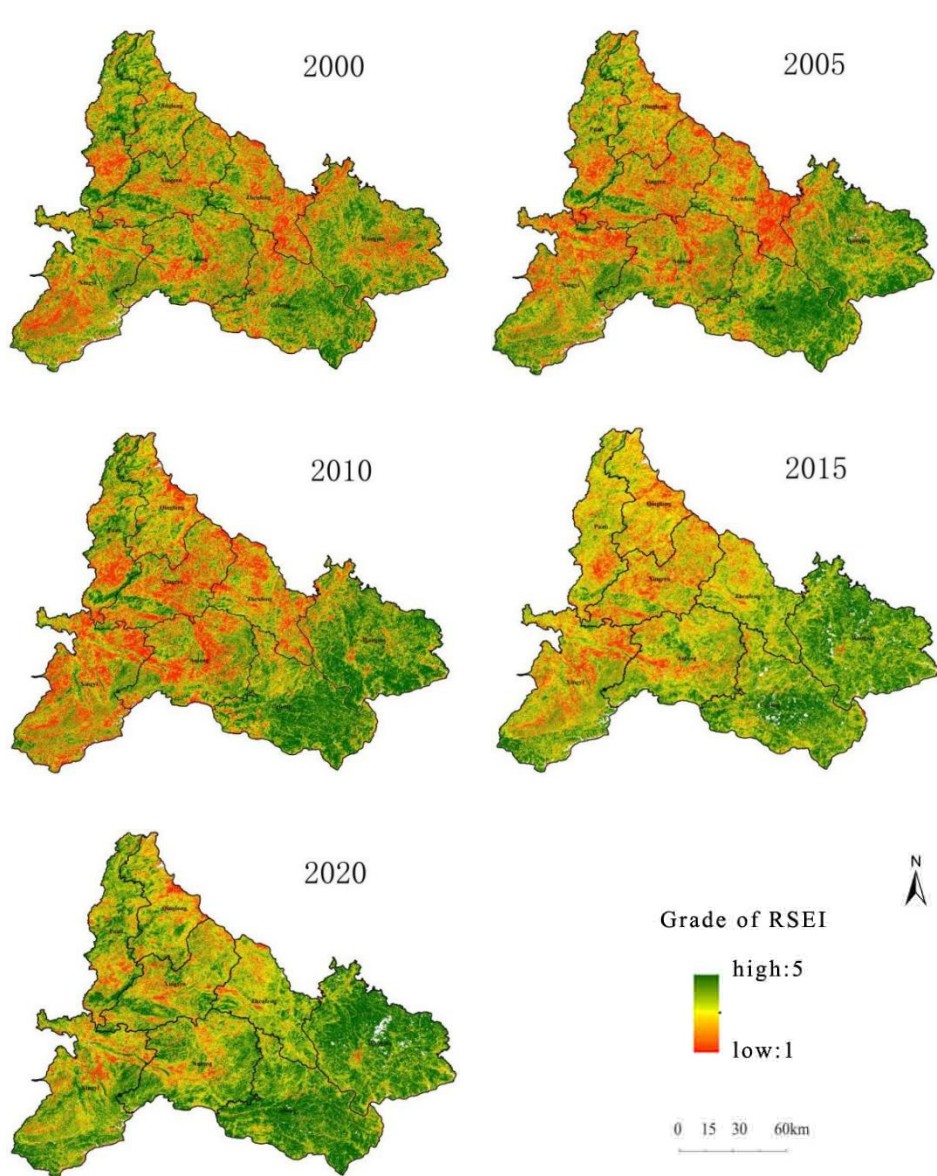

**Figure A1.** Distribution of RSEI from 2000 to 2020.

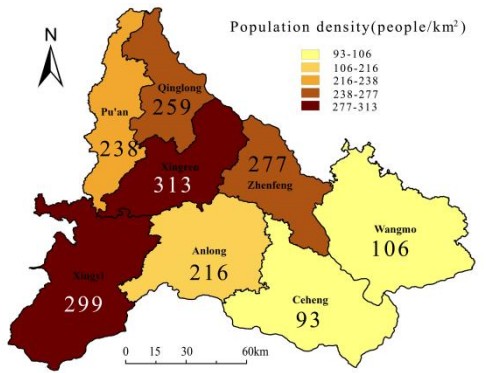

**Figure A2.** Population density of 8 counties (2016).

**Table A1.** Statistical table of changing EEQ image factors.

| Year | RSEI | MAT (°C) | ATP (mm) | Grain to Green ($10^{10}$ CNY) | Karst Rocky Desertification Restoration ($10^{10}$ CNY) |
|---|---|---|---|---|---|
| 2000 | 0.5329 | 15.67 | 1395.22 | 0.09 | |
| 2001 | | 16.35 | 1459.53 | 0.10 | |
| 2002 | 0.5439 | 16.66 | 1218.52 | 1.02 | |
| 2003 | 0.537 | 16.81 | 1233.95 | 1.73 | |
| 2004 | 0.5326 | 15.92 | 1121.14 | 1.60 | |
| 2005 | 0.5294 | 16.03 | 1259.33 | 1.68 | |
| 2006 | 0.5499 | 16.38 | 1226.83 | 1.48 | |
| 2007 | 0.5582 | 16.36 | 1428.46 | 1.37 | |
| 2008 | 0.5322 | 15.61 | 1489.99 | 1.25 | 0.44 |
| 2009 | 0.5311 | 16.61 | 1011.30 | 1.15 | 0.63 |
| 2010 | 0.5342 | 16.66 | 1291.39 | 0.78 | 0.63 |
| 2011 | 0.5322 | 15.36 | 811.22 | 0.39 | 4.25 |
| 2012 | | 15.83 | 1206.55 | 0.28 | 4.25 |
| 2013 | 0.5735 | 16.53 | 941.18 | 0.15 | 4.25 |
| 2014 | 0.5767 | 16.43 | 1540.23 | 0.88 | 4.25 |
| 2015 | 0.5628 | 16.93 | 1484.35 | 0.95 | 4.25 |
| 2016 | 0.5565 | 16.65 | 1270.80 | 0.95 | 6.25 |
| 2017 | 0.5942 | 16.46 | 1359.94 | 0.95 | 6.25 |
| 2018 | 0.6374 | 16.42 | 1309.58 | 0.95 | 6.25 |
| 2019 | | 16.80 | 1438.02 | 0.95 | 6.25 |
| 2020 | 0.6363 | 16.43 | 1489.82 | 0.95 | 6.25 |

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
