# Peer review of "The Spatial and Temporal Evolution of Ecological Environment Quality in Karst Ecologically Fragile Areas Driven by Poverty Alleviation Resettlement"

_land, doi:10.3390/land11081150_

Round 1
Reviewer 1 Report
In my opinion, this work is not about ecological quality, because the authors only work with the ecological index that was constructed with four general indicators: greenness, wetness, heat, and dryness. Through these indicators is difficult to establish a good ecological condition of ecosystems: e.g. greenness could be derived from a genetic modification and intensive (biocides applying) soja crop, and of course, this type of land use does not imply a good ecological condition of the ecosystem.
The paper describes a technical work about map/land cover analysis, but it does not establish the relationship between index results and ecological ecosystem conditions.
The paper could be re-oriented/re-writing in a direction of technical work about map-image processes.
Reviewer 2 Report
This article explores the ecological consequences of China’s poverty alleviation resettlement program (PAR). While the topic is an important one, this article is unfortunately not publishable in its current form. In addition to needing extensive editing, the authors present a biased view of PAR in the opening paragraphs of the article. For instance, the authors write in the first paragraph that the “ecological restoration effect of the karst mountainous region in the southwest China is the most outstanding.” This is more an opinion than an objective statement. Additionally, the authors speak of PAR in absolute terms of success in the introduction. Since the goal of the paper is to evaluate the effectiveness of ecological restoration using PAR, such absolute statements indicate an inherent bias in the study. Instead, PAR should be presented based on its goals and this study should be characterized as evaluating whether it has met one or more of these goals.
One immediate issue is the others use “PRA” often throughout the paper instead of “PAR.” I recognize this as a typo, but it is a frequent and consistent typo.
The authors also discuss the ecological effects of PAR, yet do not explicitly discuss how PAR was implemented in Southwest Guizhou Autonomous Prefecture, at least not until the results. In the results they discuss the relocation in the five counties on the northeast side of the study area. This really should be presented either in the introduction or the methods (study area) to show the importance of this resettlement to the ecology of the region. I think a robust discussion of the environmental pressures these regions placed on the karst environment area also important and how, beyond the indices used, there is demonstrable improvements in the region.
Overall, the remote sensing methods are sound. The authors use Google Earth Engine to calculate a number of indices (NDVI, WET, NDSI, and LST, which represent greenness, humidity, dryness, and heat, respectively, and are combined to calculate the remote sensing ecological index (RSEI). However, the authors oversell their results. In the opening sentence of the discussion, for instance, they write, “the findings verify that a reasonable reduction in human disturbance to nature is the basis of achieving harmonious development of humans and nature.” I would argue that the results show that, in the absence of direct human pressures, there are moderate increases in ecological integrity. Given the relatively short period of investigation, the study only shows the potential long-term benefits of this program.
Overall, this manuscript needs some fairly significant work before it is publishable. In addition to the English language editing, the authors need to take a far more objective approach to describing PAR. For instance, discussing it intent, how it was implemented, areas in which it was implemented, and its potential ecological benefits.
Reviewer 3 Report
The article addresses the issue of The spatial and temporal evolution of ecological environment quality in karst ecologically fragile areas driven by poverty alleviation resettlement. The topic combines the crucial issue of tackling poverty and ecologically fragile areas. However, the solution to this topic is explained very superficially. The results thus remain unsubstantiated.
The issue of PAR is also debatable. I think forcing rural farmers to live in the city is not an appropriate price to increase NDVI. It is also not possible to automatically assume that the desired succession will take place on the vacated areas. However, I understand that, for example, the level of water pollution may decrease. On the other hand, many valuable ecosystems may be dependent on traditional forms of rural farming. It would be good to compile a summary table of positive and negative effects of PAR
In addition, the authors replaced the acronym PAR with PRA e.g. between lines 93 and 108.
Equally crucial is the methodological explanation of the formulas. The methodology describes what quantities the individual indices represent, but do not substantiate them with values, and especially data sources (are these measured data? Or secondary data? What are the quantities of the units?) It would be good for the authors to document the original data entering the calculation and their resources. If there are values ​​for the formulas (eg lines 252 - 253), it is not clear how the authors arrived at these values.
In addition, the results are expected - less population = more greenery. A fundamental issue that has not been proven in the article is the quality of the new green areas.
As for the literature used, I would recommend expanding it more with non-Chinese publications focusing on succession in areas with declining human population density.
Round 2
Reviewer 1 Report
The additional explanations improve the final manuscript, and it is possible for a better understanding of the process of research and the results
Author Response
Thank you very much for your valuable comments on our article. Have a nice day.
Reviewer 2 Report
This is a resubmission of an article that explored the ecological consequences of China’s poverty alleviation resettlement program (PAR). In the original review, I noted that this was an important topic, but the article was not publishable in the current form for several reasons. Most notably, the original version did not appear to objectively evaluate PAR, rather accepted the benefits of it at face value. This was problematic because it showed a bias, perhaps unintentional, on the part of the authors.
The authors appear to have taken my (and perhaps other reviewers’) comments to heart and have significantly tamped down the praise and present PAR in a more objective light. This is a major improvement to the paper and may help alleviate any concerns that other readers may have.
My second issue was the authors’ failure to explicitly discuss the implementation of PAR in the Southwest Guizhou Autonomous Prefecture. The authors did discuss the rural population (though the numbers are from 2009), the poverty incidence rate, and the ecological fragility of the region.
The authors have added a new paragraph in the methods section specifically on PAR. This includes a discussion of how PAR in the Southwest Guizhou Autonomous Prefecture involved 1,222 villages, 74,600 households, and 338,600 individuals that were resettled to 65 sites from 2016 to 2019. These are important figures and I am happy the authors addressed this concern.
My third concern is the feeling that the authors have oversold their results. As I noted in my original comments, the results show that in the absence of direct human pressures, there are moderate increases in ecological integrity. I still think that this is all we can infer from the results.
I have read and re-read the results and, while looking at several factors, both anthropogenic and natural, I am reluctant to attribute the change over the past six years primarily to PAR. However, I think the results show that PAR may be able to deliver on what of its stated goals.
My final concern with the article is it needs extensive English editing. This remains the case. I have an advantage here in that I am evaluating the English writing proficiency of non-native speakers whereas I am a native speaker. Nonetheless, I strongly urge the authors to find an English language editor to help with this issue.
All this to say that the article has shown substantial improvement since the last version. The authors have largely addressed my concerns about potential bias—at least now it doesn’t seem as explicit. The authors also did a much better job of discussing the implementation of PAR in this region and how it could affect ecological integrity. Nonetheless, the authors still may be slightly overselling their results and I fundamentally believe this article still needs some extensive language editing to be publishable.
I commend the authors on the revisions they have made thus far.
Author Response
Dear Reviewers:
Thank you very much for your valuable comments on "Spatial and temporal evolution of ecological quality in karst ecologically fragile areas driven by poverty alleviation and relocation" (ID.Land-1776078). These comments are very valuable for us to revise and improve the article, and have important guidance for our research. We have carefully studied these comments and made revisions and improvements. And we have made new adjustments to the overall English editing of the article (Appendix 3), We have discussed the rural population and the poverty incidence rate using the number from 2015.
Reviewer 3 Report
Dear authors, thank you for the modifications and also for accepting and incorporating the comments. Thank you for explaining the PAR mechanism.
I still insist on one comment - the lack of a larger number of foreign publications. I think the main problem with PAR is its unsustainability. Farmers who forcibly relocate to cities will have problems with lifestyle changes and retraining. On the contrary, abandoned areas can of course solve the succession. On the other hand, native plant and animal species do not always form biodiversity and problems with invasive species can occur. The global trend to help in rural areas is to promote traditional and sustainable farming, nature-friendly measures, rather than relocating the rural population to cities. Rather, the aim should be to maintain local biodiversity and reduce poverty through the sustainable and close-to-nature use of local resources.
That is why I lack more foreign resources, including the international policies of the CBD, FAO, UNDEP, UNEP, etc. I think that the promotion of biodiversity through PAR is in direct conflict with the philosophy of these global trends. And that is what needs to be discussed and compared.
Otherwise, the research methodology itself is described in a sufficient way.
